# Performance of Chlorella Vulgaris Exposed to Heavy Metal Mixtures: Linking Measured Endpoints and Mechanisms

**DOI:** 10.3390/ijerph18031037

**Published:** 2021-01-25

**Authors:** Nora Expósito, Roberta Carafa, Vikas Kumar, Jordi Sierra, Marta Schuhmacher, Gemma Giménez Papiol

**Affiliations:** 1Environmental Engineering Laboratory, Chemical Engineering Department, Rovira i Virgili University, Av. Països Catalans 26, 43007 Tarragona, Spain; nora.exposito@urv.cat (N.E.); roberta.carafa@fundacio.urv.cat (R.C.); marta.schuhmacher@urv.cat (M.S.); gemma.gimenez_papiol@sorbonne-universite.fr (G.G.P.); 2Laboratory of Toxicology and Environmental Health, School of Medicine, IISPV, Rovira i Vir-gili University, Sant Llorenç 21, 43201 Reus, Spain; sierra@ub.edu; 3Faculty of Pharmacy, Barcelona University, Avda Joan XXIII s/n, 08028 Barcelona, Spain

**Keywords:** toxicity, heavy metal mixtures, metals speciation, microalgae, PAM

## Abstract

Microalgae growth inhibition assays are candidates for referent ecotoxicology as a fundamental part of the strategy to reduce the use of fish and other animal models in aquatic toxicology. In the present work, the performance of *Chlorella vulgaris* exposed to heavy metals following standardized growth and photosynthesis inhibition assays was assessed in two different scenarios: (1) dilutions of single heavy metals and (2) an artificial mixture of heavy metals at similar levels as those found in natural rivers. Chemical speciation of heavy metals was estimated with Visual MINTEQ software; free heavy metal ion concentrations were used as input data, together with microalgae growth and photosynthesis inhibition, to compare different effects and explain possible toxicity mechanisms. The final goal was to assess the suitability of the ecotoxicological test based on the growth and photosynthesis inhibition of microalgae cultures, supported by mathematic models for regulatory and decision-making purposes. The *C. vulgaris* algae growth inhibition test was more sensitive for As, Zn, and Pb exposure whereas the photosynthesis inhibition test was more sensitive for Cu and Ni exposure. The effects on growth and photosynthesis were not related. *C. vulgaris* evidenced the formation of mucilaginous aggregations at lower copper concentrations. We found that the toxicity of a given heavy metal is not only determined by its chemical speciation; other chemical compounds (as nutrient loads) and biological interactions play an important role in the final toxicity. Predictive mixture effect models tend to overestimate the effects of metal mixtures in *C. vulgaris* for both growth and photosynthesis inhibition tests. Growth and photosynthesis inhibition tests give complementary information, and both are a fast, cheap, and sensitive alternative to animal testing. More research is needed to solve the challenge of complex pollutant mixtures as they are present in natural environments, where microalgae-based assays can be suitable monitoring tools for pollution management and regulatory purposes.

## 1. Introduction

Several environmental regulations rely on ecotoxicological data for the assessment and management of chemical contaminants. In Europe, the Water Framework Directive (Directive 2000/60/EC [1]) defines the protection of European waters (surface continental waters, transition waters, coastal waters, and subterranean waters) and regulation, management, monitoring, assessment, etc. rely, among others, on the ecotoxicological test. 

Ecotoxicology has a holistic approach to the study of a toxic effect on the ecosystem. It has gained a fundamental role in the assessment and management of environmental risks [2]. In aquatic ecotoxicology, the tested compartments typically belong to three basic trophic levels: microalgae (at the bottom of the trophic chain), zooplankton (the intermediate link), and fish (top trophic level) [3]. Depending on the concentration of the compound and the exposure time, the tests are able to assess chronic or acute toxicity. 

The OECD’s (Organisation for Economic Co-operation and Development) revised version of acute toxicity tests [3,4] states that fish-based tests are only to be performed at one concentration of toxic compound: the lowest toxic concentration for microalgae or zooplankton. It is based on results showing a higher sensitivity of microalgae and zooplankton [5,6,7] and the European requirement of reducing the use of animals in laboratory assays [8,9]. Consequently, zooplankton- and microalgae-based tests (growth and photosynthetic activity the parameters mostly measured) are a fundamental part of the strategy of reducing the use of fish, including fish larvae and other animal models, in aquatic toxicology, and several standardized protocols on microalgae growth inhibition assays have been published [3,10,11,12]. 

However, for other toxicity assays involving sublethal effects, such as neurobehavioral disorder (affecting reproductivity capacity), the use of zebrafish as the organism model is more feasible [13]. Likewise, for studies of excretion kinetics and preferential pathways of metal nanoparticles with possible applications in new treatment for neoplasms or neurological disorders, not only is a zebrafish model suitable [14,15], but also murine animal models [16]. 

In microalgae growth inhibition assays, the half-maximal effective concentration (EC50) of a compound is a toxicity endpoint based on algal population parameters such as abundance, growth rate, and biomass. Standardized toxicity tests usually last between 2 to 120 h and examine the effects upon multiple generations of an algal population, thus they should not be considered acute toxicity tests as in animal tests [17].

Heavy metals are considered essential or toxic depending on their role in microalgae physiology and their concentration. Generally, Cu and Zn are a priori essential or micro-nutrients for many microalgae species, Ni is essential for diatoms, and As and Pb are non-essential [18,19]. Some studies have shown greater metal effects on microalgae growth at 48 h in microalgae cultures and species showing high growth rates [20]; for this reason, the length of the assays can be reduced in order to save costs, get faster assessments, and avoid misleading results about toxicity with the development of microalgae detoxification mechanisms [17,21].

Several studies showed that heavy metals inhibit microalgae photosynthesis (e.g., [22,23]). Photosynthesis efficiency, also called photosystem II (PSII) quantum yield or photosynthetic yield, is the fraction of the calories of radiation absorbed, which are stored as calories of chemical potential. Photosynthesis inhibition measurement by pulse-amplitude-modulation (PAM) fluorometry and the saturation pulse method is based on the fact that light energy captured by photosynthetic pigments can be either (i) used to drive photosynthesis, (ii) be dissipated as heat, or (iii) emitted as light (fluorescence). An increase in the yield of one process is directly linked to a decrease in the other two. It is therefore possible to measure a change in the efficiency of photochemical processes by measuring the yield of chlorophyll (or other pigments) fluorescence [24,25,26]. The PAM fluorimeter operates with a specific modulation of the measuring light signal that allows for the detection only of fluorescence excited by the measuring light. This method has been proven to be very efficient for the detection of effects of metals in microalgae as well as a very wide range of other toxicants (for a review, see [27]). PAM fluorometry applied to assessing short- and long-term toxic effects on microalgae provides several advantages: (1) a combined and rapid evaluation of several functional parameters in parallel, (2) screening of trends over time, (3) observing effects in replication, and (4) being non-destructive.

The sensitivity to a toxic compound in microalgae is species-dependent [28]. Therefore, few freshwater microalgae species are used as model species, such *Chlorella vulgaris*. This species fulfils three basic requirements: It is a single cell that does not form aggregates in culture and test conditions, it is easy to maintain in laboratory conditions, and it is very sensitive to contaminants. Additionally, the combined toxicity of a mixture of compounds depends on complex interactions between the test organism, the contaminants, and other compounds (i.e., nutrients) present in the sample. 

Many mathematical models have been developed for the analysis of toxicological data and chemical speciation, and eventually for understanding the mechanisms of action and supporting pollution management, regulation, and related decision-making. The most common toxicological endpoints used in algal growth and photosynthesis inhibition tests are cell density and cell photosynthetic yield at the end of the exposure period. A mathematical description of the relationship between the pollutant concentration (dose) and the final cell density and cell photosynthetic yield (response) is useful to interpolate, extrapolate, or derive metrics such as EC50, other effective concentrations (ECx), or maximum effects of toxicity. In most cases, a single inflection is observed, which can be modeled using a classical Hill model [29]. In other cases, the laboratory results show an initial stimulatory effect (hormesis), a full inhibition with two inflection points (bi-phasic curve), or a combination of these features, which requires other mathematic descriptions [30]. The toxicity of heavy metals depends on their bioavailability, which is influenced by their chemical speciation. Heavy metal speciation in aquatic environments depends on the conductivity and pH of the medium, as well as the other dissolved compounds (i.e., their chemical speciation and their concentration). 

The Eh-pH diagram of The National Institute of Advance Industrial Science and Technology of Japan [31], based on the pH range of the culture medium, is an extensively used model for determining the chemical speciation of heavy metal. More elaborated models such as Visual MINTEQ [32] also take into account the temperature, the concentration, and the chemical speciation of other chemicals in the medium. 

Heavy metals are present in nature rarely as a single compound; instead, a variety of mixtures is commonly present [33]. How to deal with the toxicity of mixtures is still an open challenge [34]. This work also analyzes the effects of some selected metal mixtures, based on some of the most detected heavy metals in the environment. 

The aim of the present work is to assess the suitability of regulatory and decision-making resolutions obtained with the ecotoxicological tests based on the growth and photosynthesis inhibition in microalgae cultures exposed to heavy metal mixtures at environmentally relevant concentrations. A combination of calculated EC50 plus metal bioavailability based on mathematical models was used. The measured effects were then compared to the prediction of the concentration addition (CA) [35]) and independent action (IA) [36,37] mixture models. Moreover, the obtained data gave the opportunity to study the differences in toxicity of the selected heavy metals in microalgae regarding the specific mode of action (MoA) and bioavailability.

## 2. Materials and Methods

### 2.1. Microalgae Culture and Quantification

*Chlorella vulgaris* (Chlorophyta, strain SAG 211_11b) was cultured in nutrient-enriched modified Bold’s Basal Medium (BBM) non-autoclaved medium [38], pH 6.8 ± 0.1, natural light (maximum irradiance 4180 μE m^−2^ s^−1^, average day length 14L:10D), and a natural temperature range from 23.7 to 39.8 °C.

The growth rate was calculated as follows:(1)$$U=(\ln\left(\frac{N1}{N2}\right))/t$$
where *U* is the growth rate, *N*1 is the cell density at the beginning of the test (cells mL^−1^), *N*2 is the cell density at the end of the test (cells mL^−1^), and *t* is the time lapse (in days). Microalgae cells were counted on a Neubauer chamber at the light microscope.

### 2.2. Reagents and Sample Preparation

All reagents were from Sigma AldrichTM (Saint Louis, MO, USA) and high purity. Single dilutions of metals were prepared in distilled water in previously cleaned glass material using previously calibrated automatic pipettes (Table 1). The concentration range of single dilutions of heavy metals in the test tubes included concentrations detected in natural samples from Catalan rivers [39,40] and the US Environmental Protection Agency screening benchmark for chronic effects [41]. The artificial mixture of heavy metals was prepared with single dilutions of each heavy metal in distilled water.

### 2.3. Ecotoxicologic Growth Inhibition Assays

Ecotoxicological assays of growth inhibition followed the conditions used for *R. subcapitata* in the paper by Expósito et al. [42]. *Chlorella vulgaris* cultures at exponential growth phase and a climatic chamber (KBWF 240, Binder GmbH, Tuttlingen, Germany) at constant 174.6 μE m^−2^ s^−1^ light and 32.4 °C were used. Initially, 190 cells µL^−1^ were distributed to glass test tubes containing BBM culture medium, plus 100 µL of single heavy metal dilution, an artificial mixture of heavy metals, or distilled water (control groups); the final volume was 2.1 mL per test tube. Five replicates were prepared per single heavy metal dilution and artificial mixture of heavy metals. At the end of the assay, 48 h later, iodine lugol was added in order to stop cell growth. Cell density was checked using a Neubauer chamber under a light microscope. Pictures of cell morphology were taken with a Nikon TE2000E microscope equipped with a camera and managed with NIS-Elements light software (Nikon Corporation, Tokyo, Japan).

The growth rate of controls was the quality control of the results: If it was equal to or higher than the specific growth rate of the microalgae culture in exponential phase, the results were accepted.

The performance of *C. vulgaris* was assessed in two different scenarios: (1) single dilutions of heavy metals and (2) an artificial mixture of heavy metals.

### 2.4. Photosynthesis Efficiency Inhibition Assays

Photosynthesis inhibition was studied in acute exposure and after 24 h exposure and incubation in the darkness in a climatic chamber (KBWF 240, Binder GmbH, Tuttlingen, Germany) at 25 °C and 75% humidity. Microalgae cultures were used when they reached the exponential growth stage. The test was performed on black 96-well flat-bottom microplates; each well contained 400 µL of culture medium (initial density of 3.3–3.8 × 10^6^ cells mL^−1^) plus 4 μL of single heavy metal dilution or distilled water (control groups). 

The effect of 12 concentrations of single heavy metals (except 6 in the case of Pb) and 7 concentrations of metal mixtures (Table 1) ranging from 0.1 µM to 10 mM was measured with a MAXI Imaging-PAM instrument (Heinz Walz GmbH, Germany) equipped with LED lights with a wavelength of 470 nm.

After 5 min dark adaption, the minimum fluorescence yield F0 was measured on a non-actinic measuring light (ML) with low intensity (3 µmol quanta m^−2^s^−1^ PAR (frequency 8 Hz, modulated pulses of 100 µs). The gain was adjusted in order to get a fluorescence signal in the range of 0.150–0.200 units. The saturation pulse was 2700 µmol quanta m^−2^ s^−1^ Photosynthetic Active Radiation (PAR) and lasted for 800 ms. The maximal level of fluorescence when all PS II centers were closed (Fm) was measured after dark adaptation. The unquenched variable fluorescence (Fv) was calculated as the difference between Fm and F0. Maximal PS II quantum yield (Max YII) was determined after dark adaptation and was calculated as Max YII = Fv/Fm. Four saturation pulses were applied for 100 s each, maintaining the sample in the dark phase to check homogeneity of the replicates. The test substances were added during a short break after pulse 4, maintaining the dark phase and starting from lower concentrations; the time delay between the first and last addition was about 200 s. The dark phase was maintained during 3 saturation pulses after the addition of the test substances in order to check the early effects on PS II not related to photosynthetic processes. Afterwards, actinic light was applied at two PAR intensities: 83 and (after 1000 s) 283 µmol quanta m^−2^ s^−1^, and additional saturating pulses were given at intervals of 100 sec to measure the maximal level of fluorescence under illuminated conditions (Fm′). The transient fluorescence (Ft) was monitored during the entire duration of the test (2000 s). The effective PS II quantum yield (YII) was calculated by the formula YII = (Fm’ − F)/Fm’, where F is the level of fluorescence immediately before the saturation pulse (3 s average). After the acute test the samples were incubated 24 h in the dark and afterwards the same measurement protocol was repeated.

### 2.5. Data Modeling

The heavy metal concentration and percentage of growth inhibition of microalgae cultures, obtained from tests with single heavy metal dilutions, were the input data for Matlab R2017a software (KTH Royal Institute of Technology, Stockholm, Sweden). All data were log-transformed prior to data analysis to ensure the normality of the data and homogeneity of variances. The results were fitted using the hormetic concentration-response model by Yoshimasu et al. [30] implemented in Matlab R2017a. Concentration-response graphs were also plotted in Matlab R2017a. Matlab software suggests the best model that fits the obtained results with single dilutions of heavy metals. Values of EC50 and expected growth inhibition from single and artificial metal mixtures of contaminants were calculated using this software, interpolating or extrapolating the missing information. Mixture models were based in the concentration addition (CA) [35] and independent action (IA) hypothesis [36,37]. In the case of CA, the following equation is applied:(2)$$ECx_{Mix}={\left(\overset{n}{\underset{i=1}{\sum }}\frac{p_{i}}{ECx_{i}}\right)}^{-1}$$
where *n* denotes the number of mixture components, *p_i_* is the relative fraction of chemical I in the mixture, and *x* is a common effect level, which is provoked by an exposure to a single substance or mixture concentration *ECx_Mix_* resp. *ECx_i_*.

In the case of IA, the following equation is applied:(3)$$E(c_{Mix})=1-\overset{n}{\underset{i=1}{\prod }}(1-E(c_{i}))$$

The individual effects of mixture constituents *E* (*c_i_*) can be calculated from the concentration response functions *F_i_* determined for single substances: *E*(*c_i_*) = *Fi* (*c_i_*). Again, the individual concentrations *ci* can be expressed as relative proportions *p_i_* of the total concentration *c_mix_*, and under the condition that the total effect *E*(*c_mix_*) equals *x%*, *c_mix_* is defined as *ECx_mix_*. Thus, by substitution we can transform Equation (3) into:(4)$$x\%=1-\overset{n}{\underset{i=1}{\prod }}(F_{i}\left(p_{i}(ECx_{mix}\right)))$$
which implicitly provides a prediction of effect concentrations of a mixture under the hypothesis of IA.

Metal speciation was simulated by Visual MINTEQ 3.1 (KTH Royal Institute of Technology, Stockholm, Sweden) [32]. This model is a chemical equilibrium model for the calculation of metal speciation, solubility equilibria, and sorption in water. The heavy metal concentration can be total (i.e., assuming 100% bioavailability, data in Table 1) or corrected with the percentage of bioavailable chemical species, estimated with Visual MINTEQ.

Data on and heavy metal concentrations in single dilutions and in artificial mixtures were the input data for determining the chemical speciation of each heavy metal in the test tubes, based on mathematic models (Appendix A). The chemical speciation of each heavy metal obtained with the Eh-pH diagram [31] was compared to the chemical speciation obtained with Visual MINTEQ [32].

### 2.6. Statistical Analysis

The statistical analysis was performed with XLSTAT software (Addinsoft, New York, NY, USA). To avoid the bias of a small number of samples and verify data normality of the growth variable before ANOVA analysis, three normality tests were applied: Lilliefors, Shapiro–Wilk, Jarque–Bera and Anderson–Darling. Significant differences were determined with ANOVA and Tukey and Dunnett’s multiple comparisons test (α = 0.05) followed by a verification of the standardized residues at 95%. 

## 3. Results and Discussion

The specific growth rate in the culture for *C. vulgaris* averaged 0.55 divisions per day, the average growth rate in control groups exposed to single dilutions of As, Pb, and Ni was 1.30 divisions per day, in control groups exposed to single dilutions of Cu and Zn it was 0.93 divisions per day, and the average growth rate of the controls exposed at combined metals test was 1.19 divisions per day. These growth rates were within the range of the growth of *C. vulgaris* in the exponential stage for cultures kept at the established culturing conditions; consequently, the results obtained were suitable for ecotoxicological assessment according to the standard procedures.

The use of a culture medium as the exposure solution rather than natural water provides an unnatural level of nutrients that is not found in natural waters. Nevertheless, the culture medium was selected for three reasons: first, in order to be sure that the detected toxicity was due to the metals or the natural sample and not to other unexpected or undetected compounds in the natural water. Second, the composition of the exposure solution was needed for the speciation/bioavailability models, the more information the better the speciation prediction would be, and culture media have a known composition. Third, the test targeted a clear toxic effect, i.e., an effect that can be detected in an optimum growth scenario such as the exponential growth stage with enough nutrients in the solution to keep on growing.

A colorless, mucilaginous matrix embedding pigmented *C. vulgaris* cells was observed (Figure 1) when exposed to low Cu (2.5–5.0 µg L^−1^) concentrations. It did not appear at higher concentrations of Cu, nor in any of the concentrations tested for the other heavy metals or the artificial mixture of heavy metals. A similar mucus was observed in *Raphidocelis subcapitata* exposed to Cu at similar concentrations, and in Zn at low concentrations [42]. The mucilage impaired the quantification of *C. vulgaris* by direct count and biases indirect quantifications based on the assumption that this species is always single suspended cells in culturing and test conditions. The external mucilage could be a defense mechanism against Cu toxicity by adsorption of the heavy metal ion to the mucilage and preventing its entrance to the cell. Secretion of metal-binding organic compounds (exopolysaccharides and exoproteins) to the surrounding environment has been described in other microalgae species and suggested as a mechanism for reducing the heavy metal toxicity by avoiding any interaction between metal cations and the microalgae cell [21]. The metal cation might form complexes with negatively charged residues of the organic compounds (pyruvate, succinate, acetate, phosphate) with a reduced bioavailability [43]. 

### 3.1. Heavy Metal Toxicity and Speciation

Heavy metal cations are considered the most toxic chemical species; nevertheless, the bioavailability of heavy metal-EDTA (Ethylenediaminetetraacetic acid) complexes should not be overlooked. Inorganic arsenic compounds, which are anions, are more toxic than organic compounds, and trivalent species (As III) are more toxic than pentavalent species (As V) [19]. As III act as a cross-linking agent by binding up to three monothiol molecules, such as the antioxidant GSH (glutathione), and this arsenic–protein binding often triggers cellular responses such as oxidative stress [44,45,46]. 

Copper, as ionic form Cu^2+^ disrupts many microalgae metabolic pathways, such as photosynthesis, respiration, ATP (Adenosine triphosphate) production, and pigment synthesis, as well as inhibits cell division [47]. According to Stauber and Florence [48], copper binds rapidly to many non-specific sites on the cell membrane, including carboxylic, sulfhydryl, and phosphate groups, and specifically to copper transport sites. Once internalized, it can oxidize thiol groups in the cytoplasm, leading to a lowering of the ratio of reduced to oxidized glutathione, which in turn affects spindle formation and cell division. 

Nickel, as ionic form Ni^2+^, is an essential metal in the cellular physiology of some eukaryotes and prokaryotes, including microalgae [49,50]. However, at higher concentrations, this metal can diminish growth [51], decrease the lipid content of autotrophic systems [52], and directly affect their photosynthetic system [53,54]. Nickel toxicity can also be related to its ability to replace essential metals in the metalloenzymes, which results in the disruption of metabolic pathways [18].

Lead (Pb) metal ions are able to replace other bivalent cations such as calcium, magnesium, and iron, and monovalent cations such as sodium, which ultimately disturbs the biological metabolism of the cell [55]. 

Zinc, as ionic form Zn^2+^, is a cofactor for enzymes participating in CO_2_ fixation (i.e., carbonic anhydrase), DNA transcription (i.e., RNA polymerase), and phosphorus acquisition (i.e., alkaline phosphatase) [56]. According to Ouyang et al. [23], Zn promotes the quantum yield of PSII, one of the components of chloroplasts and a clue to photosynthesis. 

The effect of ethylenediaminetetraacetic acid (EDTA) on heavy metal toxicity is heavy metal- and EDTA concentration-dependent. The general assumption that heavy metals are less bioavailable when they form chemical complexes with ligands such as humic acids or synthetic compounds (i.e., EDTA) was contradicted long ago. For instance, Tubbing et al. [57] evidenced that Cu is biologically available for *Selenastrum capricornutum* (synonym of *R. subcapitata*) when complexed with EDTA, and Vassil et al. [58] found a 78-fold higher Pb concentration when exposed to EDTA in tissues of *Brassica juncea*. Opposite results were found by Franklin et al. [59] and Ma et al. [60]: The addition of 34 µM EDTA into a *Scenedesmus subspicatus* culture enabled a 55% reduction in growth inhibition exerted by 40 µM Cu. In aquatic environments, free metal ions are in equilibrium with metals bound to organic and inorganic compounds [21]; both can be bioavailable and cause toxicity, and therefore they should be studied individually.

According to the Eh-pH diagram, the chemical species of As, Cu, Ni, Pb, and Zn were inorganic arsenate (HAsO_4_^2−^), copper (II) (Cu^2+^), nickel (II) (Ni^2+^), lead (II) hydroxide (PbOH^+^), and zinc (II) (Zn^2+^), respectively, which are assumed to be the most toxic chemical species because of their bioavailability. Based on these results, 100% bioavailability could be assumed. 

According to the Visual MINTEQ model (Appendix A), the chemical species of As, Cu, Ni, Pb, and Zn were more varied, with a smaller percentage of the most toxic species considered a priori (HAsO_4_^2−^, Cu^2+^, Ni^2+^, PbOH^+^ and Zn^2+^) and showing a strong interaction with the EDTA present in the BBM culture medium. 

Heavy metals interact with other chemicals present in a BBM culture medium. Visual MINTEQ takes these interactions into account when estimating the chemical speciation. In these conditions, all metals formed complexes with EDTA (Appendix A). In the growth inhibition tests with single heavy metal dilutions, EDTA-heavy metal complexes ranged from 16.8% (for ZnEDTA^2–^) to 99.9% (for CuEDTA^2–^ and NiEDTA^2–^); in these cases, the free ion concentrations were dramatically reduced compared to the assumption based on the Eh-pH diagram. The percentage of free ions also depended on the concentration of the source compound in the tested dilutions, i.e., the percentage increased with higher concentrations of the compound.

As the medium contains EDTA (8.5 mg L^−1^) and other components that can chelate metal ions, only metal species, which were bioavailable, as well as effective concentrations and free ions were considered for the EC50 calculations. The range of concentrations used for photosynthesis inhibition tests was wider (Table 1, Appendix A). For Cu all species were considered except CuEDTA^3−^ and CuHPO_4_ aq. Serra et al. [61] found that Cu toxicity in fluvial periphyton was reduced 1.6 times in a test performed with a high soluble reactive phosphorus (SRP) concentration (~50 µM). Since it was not expected that the microalgae would suffer from phosphorus (P) limitation during exposure under laboratory conditions (low SRP test ~5 µM), the study supported the hypothesis that the P-Cu interaction in the media leads to a reduction in Cu bioavailability. For this reason, we excluded from our calculation of EC50s the fraction of P-Cu complexes. Dissolved Zn^2+^ was the reactive species considered for Zn, and Ni^2+^ was the reactive species considered for Ni. Cu, Zn, and Ni showed the most complex behaviors: (a) at lower concentration, they had a high sequestration rate by EDTA in common and decreasing bioavailability, (b) whereas at higher metal concentration, the EDTA was saturated and the metal became almost totally bioavailable. Arsenic was dissolved and bioavailable as anions HAsO_4_^2−^ (46.30%) and H_2_AsO_4_^−^ (53.70%), and the two species had about the same concentration in the medium. Lead (Pb) was complexed by EDTA as PbEDTA^2−^ with a very high percentage (>99%), and only a small fraction was bioavailable as Pb^2+^.

### 3.2. Ecotoxicological Growth Inhibition Tests

The calculated EC50 for growth inhibition tests using 100% bioavailability and taking into account the chemical species estimated by Visual MINTEQ (Table 2) was based on Matlab concentration-response curves (Appendix A). Parameters and goodness-of-fit coefficients are indicated in Appendix A. *Chlorella vulgaris* was less sensitive than *R. subcapitata* exposed to the same heavy metal concentrations in the same conditions [42]. The 48 h EC50 indicated that for this type of test *C. vulgaris* was more sensitive to As (36, 22 µg L^−1^ for both speciation models), followed by Zn (437.6 µg L^−1^ based on the chemical speciation estimated with Eh-pH diagram and 2.984 µg L^−1^ based on chemical speciation estimated with Visual MINTEQ).

Yan [44] and Guo et al. [45] hypothesized that HAsO_4_^2−^, which is a molecular analogue of phosphate (HPO_4_^2−^), can compete for phosphate anion transporters (transporter proteins). Once in the cell, As (V) can be readily converted to As (III), the more toxic of the two forms. Trivalent arsenicals bind to thiols that are contained in numerous intracellular and cell-surface functional proteins.

Zinc essentiality can explain the higher concentrations needed for achieving growth inhibition; however, the zinc toxicity is related to the disruption of calcium uptake, another essential cation necessary for the Ca-ATPase activity in cell division [62]. 

The calculated 48 h EC50 by Eh-pH diagram for As and Zn of 36.2 and 437.6 µg L^−1^, respectively (Table 2 and Appendix A), were close to the US EPA screening benchmark [41] (Table 1), but the values estimated by Visual MINTEQ for Zn were far below the US EPA benchmark (Table 1 and Table 2). 

*C. vulgaris* exposed to Cu, Ni, and Pb gave a tendency to negative impact on growth but it was not possible to obtain a clear concentration-response curve with the concentrations used. In Table 2 some selected lower-level effective concentrations are also reported; this information was useful when comparing to photosynthesis inhibition tests (Table 3). 

The calculated 48 h EC50 for Cu, Pb, and Ni based on the chemical speciation estimated with the Eh-pH diagram and hormetic model were very high and out of the range of tested concentration (Table 2 and Appendix A); nevertheless, the calculated 48 h EC50 for Pb estimated by Visual MINTEQ was slightly below the US EPA screening benchmark (Table 1 and Table 2). The toxicity of heavy metals has been studied and determined by many authors for different microalgae species, including *R. subcapitata*, and 100% bioavailability was assumed, i.e., all dissolved heavy metal becomes the cation estimated by the Eh-pH diagram. Their findings are publicly available in scientific publications and toxicological databases. 

Single metallic species rarely exist in the natural environment; instead, the metal ions present in the natural environment might enter a variety of interactions with physical and chemical water parameters and microorganisms. The co-existence of a multiplicity of metal ions is considered to give rise to distinct interactive effects [54], namely, synergism, antagonism, and non-interaction. In experimental assays for an artificial mixture test using natural dissolve metals and metalloids concentration from the Puig River (Catalonia region) shown in Table 4 and Table 5, no inhibitory effect was observed because no significant differences were observed between this treatment and control (Figure 2). The experimental results suggest antagonist effects among them with possible inhibition of their individual toxic effect.

Expected growth inhibition concentrations were obtained from single metal concentration-response curves using the concentration addition (CA) and independent action (IA) models (Table 4 and Table 5). The EC50 calculated with the CA model was 160.84 µg L^−1^ (fitting parameters available in Appendix A), using dose response curves obtained with nominal concentrations assuming 100% bioavailability. Meanwhile the predicted EC50 calculated by the CA and IA models, assuming only free ions bioavailable in the mixture, were 30.610 µg L^−1^ and 34.17 µg L^−1^, respectively (Table 4, model fitting parameters available in Appendix A). The growth inhibition measured after the exposure to the experimental mixture was lower with respect to the expected growth inhibition by the CA and IA models (%). This result indicates possible antagonistic effects in the metal mixture (Table 5). 

### 3.3. Photosynthesis Efficiency Inhibition Assays

The calculated EC50 for photosynthesis inhibition tests, taking into account the chemical species estimated by Visual MINTEQ (Table 5), was based on Matlab concentration-response curves (Appendix A). The parameters and goodness-of-fit coefficients are indicated in Appendix A. The effects on photosynthetic yield YII in *C. vulgaris* were measured immediately after the addition of the metal (2000 s test) at 83 PAR (photosynthetically active radiation) and after an incubation period of 24 h at 83 and 283 PAR intensities. The results indicated that moderate and low concentrations of metals had a small stimulating effect (Appendix A). Effective nominal concentrations were quite high in most of the cases (>1000 µg L^−1^) and only copper, nickel, and zinc showed negative effects. Copper was the most effective metal (Appendix A), and in the dark phase a small amount of inhibition (10–20%) was already observed at the highest concentrations (results not shown). Light intensity affected the magnitude of the response: When more processes were activated in the cells, the effects of the metals appeared more quickly (Appendix A). Lower concentrations could affect the algae, but a quick recovery was shown. After 24 h exposure, a very steep concentration-response curve was observed. According to Visual MINTEQ, divalent metals showed interactions with EDTA present in the culture medium, with the free ion concentrations in some cases being dramatically reduced compared to the nominal concentration of metal added in each test. 

Copper has a vital function in the regulation of PSII-mediated electron transport either as a part of a polypeptide involved in electron transport, or as a stabilizer of the lipid environment close to the electron carriers of the PSII complex [63]. Nevertheless, Cu is a potent inhibitor of photosynthesis [27]. High Cu concentrations inhibit the photosynthetic electron transport rate, especially the PSII, and both Cu deficiency and Cu toxicity interfere with pigment and lipid biosynthesis and, consequently, with the chloroplast ultrastructure, thus negatively influencing the photosynthetic efficiency. Chen et al. [64] found a significant decline of Y (II) in *C. vulgaris* exposed to a concentration of ~190 µg L^−1^ Cu (II), to almost zero at a concentration more than ~254 µg L^−1^ (acute test). Xia and Tian [65] found that *Chlorella pyrenoidosa* exposed for a 12 h period to 130, 320, 640, 1.270, and 2.540 µg L^−1^ Cu exhibited a significant decrease in the amount of active PSII reaction centers per excited cross section (39.5%, 41.9%, 59.8%, 55.1%, and 61.2%, respectively). Ouyang et al. [23] reported a reduction in photosynthesis by *C. vulgaris* in the presence of 320 µg L^−1^ Cu. 

These values are about one order of magnitude lower respective to the effective concentration of the tests conducted in this study (Table 6). This result can be explained due to specific algae strain resistance and taking into account our specific culture condition of using a eutrophic or nutrient-enriched medium. Apart from affecting the photochemistry of PSII, copper also slows down the synthesis of PSII D1 protein, thereby inhibiting the recovery from photo inhibition in *C. pyrenoidosa* [66]. 

Zn^+2^ acts as a substitute of Mg, which is the central atom of chlorophyll; this mechanism leads to the breakdown of photosynthesis [67]. Early toxic effects of Zn on photosynthetic activity of the green alga *Chlorella pyrenoidosa* were previously studied by Plekhanov and Chemeris [68]; the early effect of heavy metals was manifested as a rapid (within 0.5–2 h) reduction of photo-induced oxygen release by the algal cells with concentration of 10 µM (~653.8 µg L^−1^). 

Zn at a concentration of 100 µM (~6538 µg L^−1^) reduced Fv/Fm in *C. pyrenoidosa* cells almost to zero after a 30 min incubation. These results are in accordance with our findings (Table 6). An analysis of the induction curve of the delayed chlorophyll fluorescence in *C. vulgaris* cells suggested that the early toxic effects of Zn at the above concentrations manifested itself not only in inhibited electron transport in PS II, but also in the reduced energization of photosynthetic membranes. 

The impact of sub-lethal concentrations of the heavy metals copper (Cu), zinc (Zn), and lead (Pb) on the photosynthesis of *C. vulgaris* was studied during 96 h exposure experiments by OuYang et al. [23]. It was found that under a concentration of 5 μM (~320 µg L^−1^), the quantum yield of PS II was reduced by Cu, promoted by Zn, and not affected by Pb. 

These results are also in accordance with our findings and can be explained by taking into account the maximum exposure concentrations for Zn. In fact, we measured an EC50 of 12.96 mg L^−1^ in the 24 h exposure test and we detected a hormetic effect at lower concentrations. For Pb the lack of effects is explained by the speciation as not bioavailable PbEDTA^2−^ (Table 5).

The green microalgae *A. falcatus* was highly sensitive to Ni concentrations as low as 1 μg L^−1^: Photosynthetic pigments were reduced and Ni^2+^ increased antioxidant enzyme responses [69]. In our study the EC50 of Ni (~10 mg L^−1^) was higher than the values found in the literature for single metal species. 

Acute toxicity effects of As were previously observed on the photosynthetic efficiency of diatoms and cyanobacteria, whereas green algae were less affected [70,71]. The toxicity measured in this study for as in photosynthesis inhibition assays was not significant with respect to the other heavy metals (except for Pb). 

Taking into account the listed previous studies, the justification of our results is probably linked to the specific species sensitivities in our algae strain and also to specific eutrophic medium conditions.

The results of the mixture experiments are shown in Appendix A and Table 3. In Appendix A the experimental data were plotted together with model AC and IA prediction. The EC50 values were comparable to the EC50 of single metal exposures but the concentration response curves were less steep and effective concentrations of EC1, 10, and 20 showed lower values with respect to both model predictions for 24 h tests. The CA and IA models underestimated the mixture effects for lower effective concentrations and slightly overestimated the mixture effects at higher concentrations (Appendix A). This indicates possible synergism in the metal effects, as the effects are more than additive (Table 5). For acute effects, as only copper showed measurable inhibition, the mixture curve was compared to the Cu concentration-response curve. Some synergistic effects are also shown in this case: The mixture was more toxic with respect to what could be predicted based on single metal exposure results (Appendix A).

The input data on heavy metal concentrations varied depending on the percentage of bioavailable chemical species, estimated with Visual MINTEQ speciation based on each culture medium. For acute tests only Cu showed measurable effects; the AC and IA models were not run for acute exposure.

The variability of the EC50 values between the growth and the photosynthesis inhibition tests and literature values can be explained by specific toxicity mechanisms and the difference in resistance of the specific algae strain and culture conditions. For example, our tests were performed with an excess of nutrient availability, with chelators resulting in higher EC50 values compared to if the tests had been conducted with lower nutrient concentrations with lower EC50 values (as Ware et al. [72] reported).

## 4. Conclusions

The methods proposed in this study have several advantages over other species models, such as a shorter time lapse, the possibility of miniaturizing and reducing the amount of chemicals used, and a high-throughput (more replicates, more data) considered as a good alternative to animal testing. Based on our results, the *Chlorella* algae growth inhibition test was most sensitive to As, Zn, and Pb exposure whereas the photosynthesis inhibition test was most sensitive to Cu and Ni exposure. The effects for growth and photosynthesis were not related. *Chlorella* evidenced the formation of mucilaginous aggregations at lower copper concentrations that could be a defense mechanism against Cu toxicity. The toxicity of a given heavy metal is not only determined by its chemical speciation; other chemicals (such as nutrients or chelators) and biological interaction play an important role in the final toxicity. Regarding nutrient concentrations, eutrophic media could protect against metal pollution. In our study, the metal sequestration rate by EDTA was important for metal bioavailability, however, the effects of the lack of EDTA in modified oligotrophic mediums have to be also considered to avoid biased results. The toxicity of metal mixtures cannot reliably be predicted based on the toxicity of individual metals using AC or IA models, because antagonistic or, more likely, synergic interactions may occur. These interactions are complex and affect the test in different way for different endpoints. More research is needed to face the challenge posed by pollutant mixtures as they are present in natural environments, and make microalgae-based assays suitable for pollution management and regulatory purposes.

## Figures and Tables

**Figure 1 ijerph-18-01037-f001:**
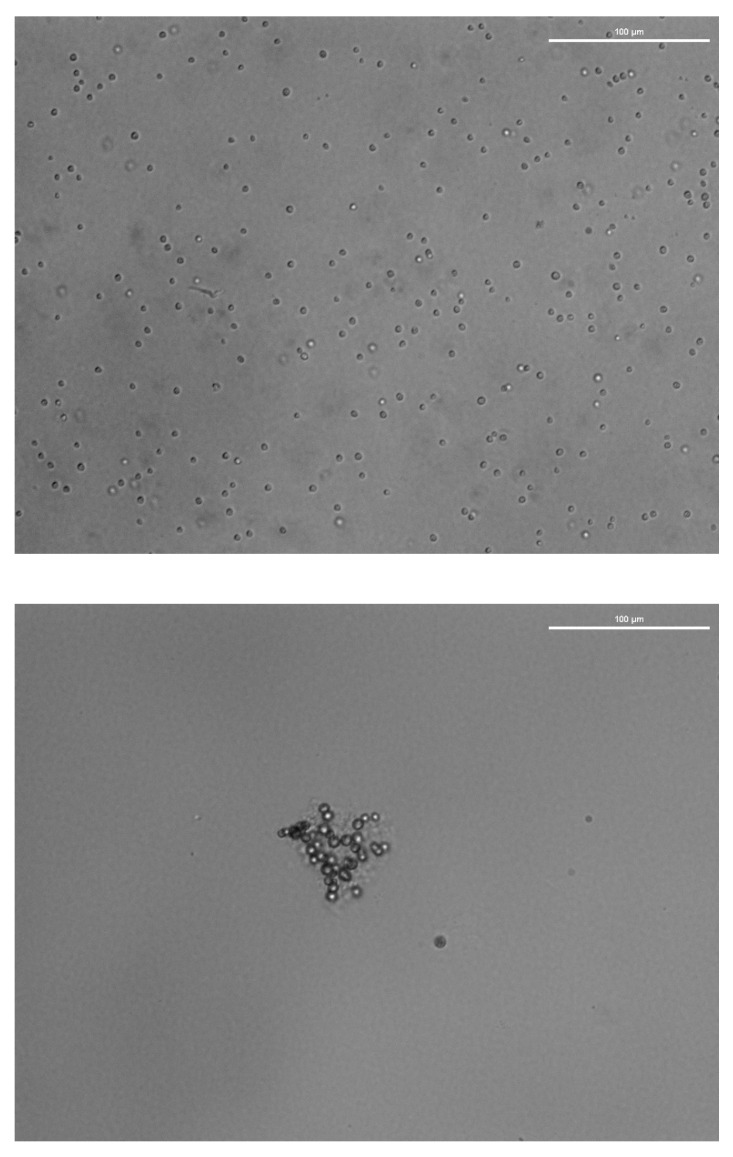
Comparison of a normal culture of *C. vulgaris*, with single, isolate, suspended cells (upper picture) and a culture exposed to low concentrations of Cu (lower picture) showing the mucilaginous matrix with cells embedded. In both cases, *C. vulgaris* cells keep their characteristic “O” shape.

**Figure 2 ijerph-18-01037-f002:**
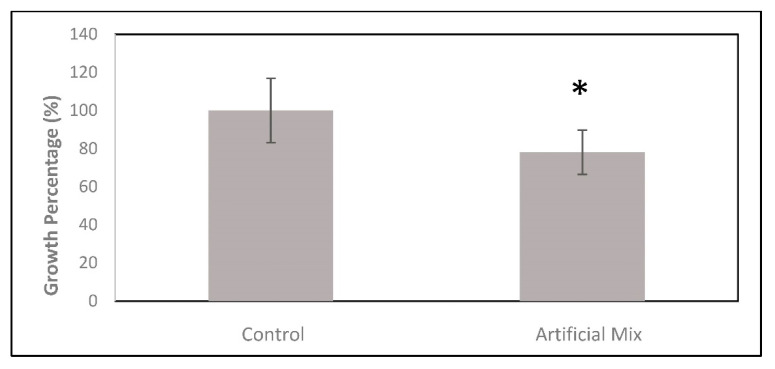
Growth of *C. vulgaris* exposed to artificial mixture of heavy metals (Artificial Mix.) (*) No significant difference to control.

**Table 1 ijerph-18-01037-t001:** Tested single heavy metal dilutions: United States Environmental Protection Agency (US EPA) benchmarks in water hardness of 100 mg L^−1^ [41], and heavy metal levels in the artificial mixture in growth (selection based on data from [39,40]) and in photosynthesis inhibition tests.

Heavy Metal	Source	Tested Concentrations in Growth Inhibition Tests (μg L^−1^)	Tested Concentrations in Photosynthesis Inhibition Tests (mg L^−1^)	US EPA Screening Benchmark (μg L^−1^) for Chronic Effects	Tested Concentrations from the Artificial Mixture in Growth Inhibition Tests (μg L^−1^)	Tested Concentrations from the Artificial Mixture in Photosynthesis Inhibition Tests (mg L^−1^)
As	HAsNa_2_O_4_	35	749.20	5 (3.1 for As V)	3.5	187.25
		17.5	374.60			93.63
		3.5	149.84			37.45
		1.7	74.92			18.73
			37.46			9.36
			29.97			3.75
			14.98			0.19
			7.49			
			3.75			
			0.75			
			0.07			
			0.01			
Cu	CuSO_4_	2.5	635.46	9	5	158.77
		5	317.73			79.39
		25	127.09			31.77
		50	63.55			15.89
			31.77			7.96
			25.42			3.19
			12.71			0.18
			6.35			
			3.18			
			0.64			
			0.06			
			0.01			
Ni	NiCl_2_	64.8	587.10	52	2.7	146.75
		13.5	293.55			73.38
		2.7	117.42			29.35
		1.3	58.71			14.68
			29.36			7.34
			23.48			2.94
			11.74			0.15
			5.87			
			2.94			
			0.59			
			0.06			
			0.01			
Pb	PbCl_2_	20	41.44	2.5	2	
		10	20.72			
		2	10.36			
		1	5.18			
			2.59			
			0.05			
Zn	ZnCl_2_	210	653.80	120	21	163.50
		105	326.90			81.78
		21	130.76			32.74
		10.5	65.38			16.40
			32.69			8.23
			26.15			3.32
			13.08			0.22
			6.54			
			3.27			
			0.65			
			0.07			
			0.01			

Note that these were the nominal concentrations of heavy metals. All concentrations referred to the single metal.

**Table 2 ijerph-18-01037-t002:** Selected effective concentrations (ECx) for 48 h exposure to single dilutions heavy metals for growth inhibition calculated with Matlab concentration-response curves.

*C. vulgaris* Test Growth Inhibition (Single Metals)			Based on the Chemical Speciation Estimated with Eh-pH Diagram	Based on the Chemical Speciation Estimated with Visual MINTEQ			
Heavy Metal	Chemical Compound Used in the Test	Heavy Metal Chemical Species	48 h EC50 (µg L^−1^)	48 h EC50 (µg L^−1^)	48 h EC1 (µg L^−1^)	48 h EC10 (µg L^−1^)	48 h EC20 (µg L^−1^)
As	HAsNa_2_O_4_	HAsO_4_^2−^ H_2_AsO_4_^−^	36.22	36.22	0.0014	0.016	0.187
Cu	CuSO_4_	CuHEDTA^−^ CuEDTA^−2^	7.36 × 10^8^	n.d.	n.d.	n.d.	n.d.
Ni	NiCl_2_	NiEDTA^2−^	1.91 × 10^7^	n.d.	n.d.	n.d.	n.d.
		NiHEDTA^−^					
Pb	PbCl_2_	Pb(CO_3_)_2_^2−^	3.66 × 10^13^	0.201	0.00001	0.00005	0.0002
		PbEDTA^−2^					
Zn	ZnCl_2_	Zn^2+^	437.62	2984	0.001	0.008	0.050
		ZnHPO_4_ (aq)					
		ZnEDTA^−2^					
		ZnHEDTA^−^					

Input data on heavy metal concentrations varies depending on the percentage of bioavailable chemical species, estimated with the Eh-pH diagram or Visual MINTEQ speciation based on each culture medium. Bioavailable species are underlined.

**Table 3 ijerph-18-01037-t003:** Maximum stimulation concentration (hEC) and selected effective concentrations (ECx, ppb) of YII in artificial heavy metal mixtures and modeled with the AC model and calculated with Matlab concentration-response curves.

		Mix	Experimental	AC Model	IA Model
*C. vulgaris* test photosynthesis inhibition (mixture metals)		Heavy Metal	Cu + Ni + Zn + As	Cu + Ni + Zn + As	Cu + Ni + Zn + As
		Chemical Compound Used in the Test	CuSO_4_ + NiCl_2_ + ZnCl_2_ + HAsNa_2_O_4_	CuSO_4_ + NiCl_2_ + ZnCl_2_ + HAsNa_2_O_4_	CuSO_4_ + NiCl_2_ + ZnCl_2_ + HAsNa_2_O_4_
		Heavy Metal Chemical Species	All Species Except Complexed with EDTA and PO_4_	All Species Except Complexed with EDTA and PO_4_	All Species Except Complexed with EDTA and PO_5_
PAR 83	acute	EC50 (μg L^−1^)	6.1 × 10^5^	5.02 × 10^4^	5.02 × 10^4^
		hEC max (μg L^−1^)	n.d.	7.90 × 10^1^	7.90 × 10^1^
		EC1 (μg L^−1^)	3.13 × 10^1^	1.75 × 10^3^	1.75 × 10^3^
		EC10 (μg L^−1^)	1.23 × 10^4^	5.46 × 10^3^	5.46 × 10^3^
		EC20 (μg L^−1^)	1.42 × 10^5^	1.23 × 10^4^	1.23 × 10^4^
	24 h	EC50 (μg L^−1^)	3.9 × 10^4^	3.66 × 10^5^	4.81 × 10^5^
		hEC max (μg L^−1^)	n.d.	n.d.	6.58 × 10^3^
		EC1 (μg L^−1^)	6.73 × 10^1^	3.16 × 10^4^	1.04 × 10^5^
		EC10 (μg L^−1^)	1.75 × 10^3^	7.90 × 10^4^	2.61 × 10^5^
		EC20 (μg L^−1^)	5.46 × 10^3^	1.50 × 10^5^	4.89 × 10^5^
PAR 263	acute	EC50 (μg L^−1^)	3.2 × 10^5^	n.a.	n.a.
		hEC max (μg L^−1^)	n.d.	n.a.	n.a.
		EC1 (μg L^−1^)	3.13 × 10^1^	n.a.	n.a.
		EC10 (μg L^−1^)	4.13 × 10^1^	n.a.	n.a.
		EC20 (μg L^−1^)	1.05 × 10^4^	n.a.	n.a.
	24 h	EC50 (μg L^−1^)	2.8 × 10^4^	8.11 × 10^4^	2.81 × 10^5^
		hEC max (μg L^−1^)	13.22	n.d.	3.51 × 10^4^
		EC1 (μg L^−1^)	2.92 × 10^2^	2.81 × 10^4^	1.08 × 10^5^
		EC10 (μg L^−1^)	1.07 × 10^3^	4.13 × 10^4^	1.32 × 10^5^
		EC20 (μg L^−1^)	2.42 × 10^3^	5.14 × 10^4^	1.57 × 10^5^

**Table 4 ijerph-18-01037-t004:** Selected expected effective concentrations (ECx) for 48 h exposure to heavy metals mixture for growth inhibition calculated with concentration-response curves using concentration addition (CA) and independent action (IA) models.

	*C. vulgaris* Modelled Growth Inhibition (Mixture Metals)			EC Mix			
				EC1 (μg L^−1^)	EC10 (μg L^−1^)	EC20 (μg L^−1^)	EC50 (μg L^−1^)
CA model growth with nominal conc	Cu + Ni + Zn + As + Pb	CuSO_4_ + NiCl_2_ + ZnCl_2_ + HAsNa_2_O_4_ + PbCl_2_	100% bioavailability	0.005	0.006	0.070	160.833
CA model growth with bioavailable conc	Zn + As + Pb	ZnCl_2_ + HAsNa_2_O_4_ + PbCl_2_	HAsO_4_^2−^ H2AsO_4_^−^ Pb^+2^ Zn^+2^	0.001	0.016	0.180	30.610
IA model growth with bioavailable conc	Zn + As + Pb	ZnCl_2_ + HAsNa_2_O_4_ + PbCl_2_	HAsO_4_^2−^ H2AsO_4_^−^ Pb^+2^ Zn^+2^	0.003	0.024	0.237	34.177

Input data on heavy metal concentrations varies depending on the percentage of bioavailable chemical species, and nominal or Visual MINTEQ speciation based on each culture medium.

**Table 5 ijerph-18-01037-t005:** Expected growth inhibitions for 48 h exposure to the artificial mixture calculated with concentration-response curves using concentration addition (CA) and independent action (IA) models.

*C. vulgaris* TestGrowth Inhibition (Mixture Metals)	Artificial Mixture of Heavy Metals					
Heavy Metal	Nominal Heavy Metal Concentration (μg L^−1^)	Measured Growth Inhibition (%)	Expected Growth Inhibition (%) Assuming 100% Bioavailability and CA Model	Free Heavy Metal Ion Concentration (μg L^−1^)	Expected Growth Inhibition (%) Assuming Bioavailability Based on Visual MINTEQ and CA Model	Expected Growth Inhibition (%) Assuming Bioavailability Based on Visual MINTEQ and IA Model
As	3.5	5.13	35.6	3.5	32.2	62.3
Cu	5			0		
Ni	2.7			0		
Pb	2			0.00018		
Zn	21			0.00441		

Input data on heavy metal concentrations varies depending on the percentage of bioavailable chemical species, and nominal or Visual MINTEQ speciation based on each culture medium.

**Table 6 ijerph-18-01037-t006:** Maximum stimulation concentration (hEC) and selected effective concentrations (ECx, ppb) of YII for Cu, Ni, and Zn, calculated with Matlab concentration-response curves for single metal tests.

*C. vulgaris* Test Photosynthesis Inhibition (Single Metals)		Heavy Metal	As	Cu	Ni	Pb	Zn
		Chemical Compound Used in the Test	HAsNa_2_O_4_	CuSO_4_	NiCl_2_	PbCl_2_	ZnCl_2_
		Heavy Metal Chemical Species	HAsO_4_^2−^	Cu^+2^, CuSO_4_, Cu(OH)_2_, CuOH^+^	Ni^+2^	PbOH^+^	Zn^+2^
PAR 83	acute	EC1 (µg L^−1^)	n.d.	1.75 × 10^3^	n.d.	n.d.	n.d.
		EC10 (µg L^−1^)	n.d.	5.46 × 10^3^	n.d.	n.d.	n.d.
		EC20 (µg L^−1^)	n.d.	1.23 × 10^4^	n.d.	n.d.	n.d.
		EC50 (µg L^−1^)	n.d.	5.02 × 10^4^	n.d.	n.d.	n.d.
		hEC max (µg L^−1^)	n.d.	7.90 × 10^1^	n.d.	n.d.	n.d.
	24 h	EC1 (µg L^−1^)	n.d.	1.71 × 10^4^	6.28 × 10^4^	n.d.	2.36 × 10^4^
		EC10 (µg L^−1^)	n.d.	2.00 × 10^4^	1.67 × 10^5^	n.d.	7.39 × 10^4^
		EC20 (µg L^−1^)	n.d.	2.13 × 10^4^	2.72 × 10^5^	n.d.	1.96 × 10^5^
		EC50 (µg L^−1^)	n.d.	2.81 × 10^4^	7.25 × 10^5^	n.d.	8.18 × 10^5^
		hEC max (µg L^−1^)	n.d.	7.57 × 10^3^	1.15	n.d.	1.02 × 10^3^
PAR 263	24 h	EC1 (µg L^−1^)	n.d.	1.71 × 10^4^	7.39 × 10^4^	n.d.	1.71 × 10^4^
		EC10 (µg L^−1^)	n.d.	2.00 × 10^4^	7.94 × 10^4^	n.d.	3.27 × 10^4^
		EC20 (µg L^−1^)	n.d.	2.13 × 10^4^	9.06 × 10^4^	n.d.	4.53 × 10^4^
		EC50 (µg L^−1^)	n.d.	2.45 × 10^4^	1.50 × 10^5^	n.d.	9.99 × 10^4^
		hEC max (µg L^−1^)	n.d.	7.57 × 10^3^	3.05 × 10^4^	n.d.	1.52 × 10^3^

Input data on heavy metal concentrations varies depending on the percentage of bioavailable chemical species, estimated with Visual MINTEQ speciation based on each culture medium.

## Data Availability

The data presented in this study are available on request from the corresponding author.

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
