# Peer review of "Performance of Chlorella Vulgaris Exposed to Heavy Metal Mixtures: Linking Measured Endpoints and Mechanisms"

_ijerph, 2021, doi:10.3390/ijerph18031037_

Round 1

Reviewer 1 Report

Manuscript Number:  IJERPH-1069094                                               

Title: Performance of Chlorella vulgaris exposed to heavy metal mixtures: linking measured endpoints and mechanisms

General comments

The paper proposed by Expósito et al. for publication in IJERPH studies the toxicity of 5 heavy metals (As, Cu, Ni, Pb, Zn) and their mixtures to the freshwater microalga Chlorella vulgaris. Effects on growth and photosynthesis were analysed. The manuscript is well written and clear, and although the toxicity of heavy metals to microalgae is not a new topic, this study provides some rather new information to the field. The work is very comprehensive and thorough work; however, there are still some issues that need to be solved and some corrections to be made before this work is suitable for publication in IJERPH.

The discussion is too extensive, and some paragraphs should be shortened and incorporated either in the introduction or after the description of results. Moreover, as this is not a new topic, the main new results and advancements for the field should be emphasised.

Comments on the manuscript

Abstract:

The last part of the abstract, from line 21 to 26, should include the main findings and the innovative information retained by the present study. Please reformulate this accordingly.

Highlights:

The obtained results should be written in the past tense.

 Introduction:

- Line 57: also refer the most recent OECD guideline 201 for microalgae testing, OECD Guidelines for the testing of chemicals – Freshwater Alga and Cyanobacteria, growth inhibition test,  from 2011.

- Lines 78, 79: use just the abbreviation “PAM”, as on line 72 the full name has already been written.

- Line 109: add a comma (,) between “…medium” and “as well as…”.

- Sentences from 117 to 121 and from 122 to 128 need to be rewritten. They are poorly written. A possible solution is to merge the information in 2 different paragraphs.

Possible solution:

“How to deal with the toxicity of heavy metal mixtures is still an open challenge. (paragraph)

The aim of the present study was to assess the suitability of regulatory and decision making resolutions obtained with the ecotoxicological tests based on growth and photosynthesis inhibition in microalgae cultures exposed to heavy metal mixtures, at environmentally relevant concentrations. A combination of calculated EC50 plus metal bioavailablity based on mathematical models was used. The measured effects were then compared with the prediction of Concentration Addition (CA, Berenbaum, 1985) and Independent action (IA, Bliss, 1939; Hewlett and Plackett, 1959) mixture models. Moreover, the obtained data gave the opportunity to study differences in toxicity of the selected heavy metals in microalgae regarding their specific Mode of Action (MoA) and bioavailability.”

Materials and methods:

Was there any chemical analysis performed, confirming the tested concentrations?

Results and discussion:

Please change the verbal forms of all observed results from present to past tense. There is a mix of both along this section.

  • Lines 285 to 290: Please remove this as it is a copy of line 2772 to 280.
  • Lines 298 to 362: These paragraphs should be shortened, and some information should be incorporated either in the introduction or after the description of results obtained for each compound. A lot of this is already repeated afterwards as well. This is more a review of what has been previously studied than the discussion of the obtained results.
  • Line 377: “Tables 2, 3 and 4”.

Conclusions:

  • As this is not a new topic, the main new results and advancements for the field should be emphasised

Author Response

Reviewer 1

Title: Performance of Chlorella vulgaris exposed to heavy metal mixtures: linking measured endpoints and mechanisms

General comments

The paper proposed by Expósito et al. for publication in IJERPH studies the toxicity of 5 heavy metals (As, Cu, Ni, Pb, Zn) and their mixtures to the freshwater microalga Chlorella vulgaris. Effects on growth and photosynthesis were analysed. The manuscript is well written and clear, and although the toxicity of heavy metals to microalgae is not a new topic, this study provides some rather new information to the field. The work is very comprehensive and thorough work; however, there are still some issues that need to be solved and some corrections to be made before this work is suitable for publication in IJERPH.

The discussion is too extensive, and some paragraphs should be shortened and incorporated either in the introduction or after the description of results. Moreover, as this is not a new topic, the main new results and advancements for the field should be emphasised.

Response: Thanks for your positive remarks and useful comments. We have considered most of your comments during this revision and it has certainly helped to improve this manuscript. All these changes can also be tracked in the revised manuscript.

Comments on the manuscript

Abstract:

The last part of the abstract, from line 21 to 26, should include the main findings and the innovative information retained by the present study. Please reformulate this accordingly.

Response: We have revised the abstract included your recommendation.

Highlights:

The obtained results should be written in the past tense.

Response: We have done extensive revision and corrected all these mistakes.

 Introduction:

- Line 57: also refer the most recent OECD guideline 201 for microalgae testing, OECD Guidelines for the testing of chemicals – Freshwater Alga and Cyanobacteria, growth inhibition test, from 2011.

Response: Thanks for this reference. We have included it.

- Lines 78, 79: use just the abbreviation “PAM”, as on line 72 the full name has already been written.

Response: Required correction made.

- Line 109: add a comma (,) between “…medium” and “as well as…”.

Response: Done

- Sentences from 117 to 121 and from 122 to 128 need to be rewritten. They are poorly written. A possible solution is to merge the information in 2 different paragraphs.

Response: Done

Possible solution:

“How to deal with the toxicity of heavy metal mixtures is still an open challenge. (paragraph)

The aim of the present study was to assess the suitability of regulatory and decision making resolutions obtained with the ecotoxicological tests based on growth and photosynthesis inhibition in microalgae cultures exposed to heavy metal mixtures, at environmentally relevant concentrations. A combination of calculated EC50 plus metal bioavailablity based on mathematical models was used. The measured effects were then compared with the prediction of Concentration Addition (CA, Berenbaum, 1985) and Independent action (IA, Bliss, 1939; Hewlett and Plackett, 1959) mixture models. Moreover, the obtained data gave the opportunity to study differences in toxicity of the selected heavy metals in microalgae regarding their specific Mode of Action (MoA) and bioavailability.”

Response: Thanks for providing this correction. We have included it.

Materials and methods:

Was there any chemical analysis performed, confirming the tested concentrations?

Response: Following the recommendations of the OECD 201 method, it was verified that at the end of the tests the concentration of metals in the culture medium (by ICP-MS) had been satisfactorily maintained within ± 20% of the nominal concentration, as Visual MINTEQ software predicted (no precipitation occurred). 

Results and discussion:

Please change the verbal forms of all observed results from present to past tense. There is a mix of both along this section.

Response: We have done extensive revision and corrected these mistakes.

  • Lines 285 to 290: Please remove this as it is a copy of line 2772 to 280. Not found ????
  • Lines 298 to 362: These paragraphs should be shortened, and some information should be incorporated either in the introduction or after the description of results obtained for each compound. A lot of this is already repeated afterwards as well. This is more a review of what has been previously studied than the discussion of the obtained results.

Response: We have changed this paragraph.

  • Line 377: “Tables 2, 3 and 4”.

Done

Conclusions:

  • As this is not a new topic, the main new results and advancements for the field should be emphasised

Response: We have revised our conclusion and highlighted our main results.

Reviewer 2 Report

The Authors have submitted a manuscript regarding the employment of microalgae for ecotoxicology investigations.

The manuscript is interesting (even if similar ones are present in literature, for example doi: 10.1007/s11434-012-5366-x), comprises several data, and the quality is enough to deserves publications. The topic addressed is of relevance for the field. Overall, this manuscript can meet the standards required by the Journal Policy after the following minors are addressed:

-many typos along the text (for example, the use of “y” instead of “and” - also in the highlights – or the complete miss of superscripts and subscripts for chemical formulae)

-Section repetitions (for example, line 272-280 and 282-290)

-In the Introduction, the Authors should briefly discuss and compare the pros/cons on the employment of Zebrafish embryos respect to microalgae (see for example, doi: 10.1021/acsabm.9b00630 and 10.3390/fishes5030023).

-From my point of view, some tables can be shifted to the SI.

-Authors should quantify the amount of metals internalized in microalgae by ICP-MS and normalize the data on the microalgae amount (for example the protein content) in order to better understand the link between toxicity and metal concentration (in microalgae)

Author Response

The Authors have submitted a manuscript regarding the employment of microalgae for ecotoxicology investigations.

The manuscript is interesting (even if similar ones are present in literature, for example doi: 10.1007/s11434-012-5366-x), comprises several data, and the quality is enough to deserves publications. The topic addressed is of relevance for the field. Overall, this manuscript can meet the standards required by the Journal Policy after the following minors are addressed:

Response: Thanks for your positive remarks and useful comments. We have considered most of your comments during this revision and it has certainly helped to improve this manuscript. All these changes can also be tracked in the revised manuscript.

-many typos along the text (for example, the use of “y” instead of “and” - also in the highlights – or the complete miss of superscripts and subscripts for chemical formulae)

Response: Further proofreading has been done and these typos are corrected.

-Section repetitions (for example, line 272-280 and 282-290)

Response: Thanks for pointing this out. We have removed repeated sentences.

-In the Introduction, the Authors should briefly discuss and compare the pros/cons on the employment of Zebrafish embryos respect to microalgae (see for example, doi: 10.1021/acsabm.9b00630 and 10.3390/fishes5030023).

Response: Though microalgae and zebrafish have a very different mechanisms and used for a different purposes. However, we have introduced a brief section in the introduction to compare them.

-From my point of view, some tables can be shifted to the SI.

Response: Tables 2-3-4 were moved to Supplementary materials.

-Authors should quantify the amount of metals internalized in microalgae by ICP-MS and normalize the data on the microalgae amount (for example the protein content) in order to better understand the link between toxicity and metal concentration (in microalgae)

Response: Internal concentrations were not analysed because other studies demonstrated that the sensitivity of algal species to metals (eg. Cu and Pb) is not related to external metal binding, intra-cellular metal concentrations nor uptake rates (Levy et al., 2008; Debelius et al.,2009).10.1016/j.aquatox.2008.06.003,  10.1016/j.ecoenv.2009.04.006